# Soil Layers Impact *Lithocarpus* Soil Microbial Composition in the Ailao Mountains Subtropical Forest, Yunnan, China

**DOI:** 10.3390/jof8090948

**Published:** 2022-09-09

**Authors:** Sijia Liu, Jiadong Wu, Haofei Wang, Anna Lukianova, Anna Tokmakova, Zhelun Jin, Shuxian Tan, Sisi Chen, Yue Wang, Yuxin Du, Konstantin A. Miroshnikov, Jianbo Xie

**Affiliations:** 1National Engineering Research Center of Tree Breeding and Ecological Restoration, Beijing Forestry University, Beijing 100083, China; 2Key Laboratory of Genetics and Breeding in Forest Trees and Ornamental Plants, College of Biological Sciences and Technology, Beijing Forestry University, Ministry of Education, Beijing 100083, China; 3The Tree and Ornamental Plant Breeding and Biotechnology Laboratory of National Forestry and Grassland Administration, Beijing Forestry University, Beijing 100083, China; 4Shemyakin-Ovchinnikov Institute of Bioorganic Chemistry, Russian Academy of Sciences, Moscow 117997, Russia

**Keywords:** co-occurrence network, dominant taxa, soil layer, soil microbiome, forest

## Abstract

Plant litter decomposition is a complex, long-term process. The decomposition of litterfall is a major process influencing nutrient balance in forest soil. The soil microbiome is exceptionally diverse and is an essential regulator of litter decomposition. However, the microbiome composition and the interaction with litterfall and soil remain poorly understood. In this study, we examined the bacterial and fungal community composition of *Lithocarpus* across soil samples from different sampling seasons. Our results displayed that the microbiome assembly along the soil layer is influenced predominantly by the soil layer rather than by the sampling season. We identified that the soil layer strongly affected network complexity and that bacterial and fungal microbiomes displayed different patterns in different soil layers. Furthermore, source tracking and community composition analysis indicated that there are significantly different between soil and litter. Moreover, our results demonstrate that few dominant taxa (2% and 4% of bacterial and fungal phylotypes) dominated in the different soil layers. *Hydnodontaceae* was identified as the most important biomarker taxa for humic fragmented litter fungal microbiome and *Nigrospora* and *Archaeorhizomycetaceae* for organic soil and the organic mineral soil layer, and the phylum of *Acidobacteria* for the bacteria microbiome. Our work provides comprehensive evidence of significant microbiome differences between soil layers and has important implications for further studying soil microbiome ecosystem functions.

## 1. Introduction

Forests cover an estimated size of 38 million square kilometers and comprise more than 3 trillion trees on earth, contributing about 90% of the primary terrestrial production [1,2]. Consequently, forests play an essential role in the global fluxes of energy and matter [1,2]. The microorganisms colonizing the soil are the most abundant and diverse life forms on earth [3,4]. Microbiomes are responsible for many vital ecosystem functions, such as the biogeochemical cycling of soil nutrients, the transformation of organic materials, the enhancement of plant productivity, and disease control [5,6,7]. The microbiome also could increase host tolerance to biotic and abiotic stress, promote stress resistance, and influence crop yield and quality [8,9]. Plants are divided into annual, biennial, and perennial plants based on their life cycle. Woody perennial plants provide several ecosystem functions such as climate regulation, nutrient cycling, and biodiversity reservoir [10]. At the same time, the associated microorganisms of forest trees contribute to the ecosystem’s evolution, metabolism, and ecology. Therefore, a complete microbiome analysis of woody plants, especially the soil microbiome, is needed for their further characterization.

Forest ecosystems provide a wide range of habitats, including the soil, litter, and habitats associated with trees (leaf, wood, bark, roots, and rhizospheres) [1]. More importantly, these habitats differ in properties such as environmental conditions and nutrient availability affecting microbial composition. Significant differences in bacterial microbiome composition have been observed in different microenvironments [11,12]. In addition, the microbiome can be variable across the plant genotypes and precipitation [13,14,15]. Notably, the soil microbiome plays a vital role in the ecosystem and is the key factor associated with soil quality, soil fertility, and productivity [16]. Soil profiles are often meters in-depth, and changes in soil structure across depth are associated with shifts in microbial communities across soil strata [17,18]. The microbial biomass, activity, and diversity are greatest in the topsoil (the top 20 cm of the soil column or less) [19]. Therefore, understanding the factors that regulate the structure and composition of the topsoil is essential.

Plant litter, especially the litter of forest trees, is the primary source of organic matter accumulation on the soil surface [1]. Plant litter decomposition is a complex, long-term process. Litter supported mainly saprotrophic taxa, while Proteobacteria and Bacteroidetes seem to be enriched in temperate forest litter than the soil regarding the bacterial microbiome [20,21]. However, the composition and co-occurrence patterns across the litter and soil remain largely unknown [22]. Previous studies have indicated that bacteria and fungi predominate in the early stages and latter stages of litter decomposition, respectively. Cover crops have been widely applied globally as a conservation agriculture practice [23]. Wang et al. reported that mulching practices increased the diversity of microorganisms and abundance of dominant species and promoted their interrelations [23]. Even though the microbiome includes diverse groups such as bacteria, fungi, and archaea, most researchers only focus on one group. Bacteria and fungi constitute the two most abundant and important groups. They have apparent developmental differences and are functionally unique but show a robust metabolic correlation [24]. Although ecological processes are ecosystem-specific, they cannot be adequately understood without considering their functioning as a whole [1]. A whole microbiome view is needed to study their composition.

Yunnan, located in southwest China, has a complex natural environment and extremely rich biological resources [25]. Here, we established and carried out a field experiment in the Ailao mountains nature reserve. The bacterial and fungal communities were investigated across 216 samples collected from different soil layers and different sampling seasons (dry and wet seasons). Our aims were to (i) identify the microbiome composition and diversity of the soil layer and the seasonal variation; (ii) determine the relative contribution of the sampling season and soil layers in shaping microbiome assemblies, and (iii) assess the co-occurrence patterns across soil layers and identify the potential source and keystone taxa of the different soil layers. Our study provides an integrated perspective on soil layer biogeography and reveals the assembly patterns and the relationship between litter and soil microbiome.

## 2. Materials and Methods

### 2.1. Description of the Sampling Site

The sampling site is located in the Xujiaba nature reserve, covering 5100 ha on the northern crest of the Ailao mountains in Jingdong Country, Yunnan Province. In the Xujiaba area, the mountainsides have been virtually stripped of their original vegetation, leaving cropland (primarily rice and maize), pastureland, and secondary forests on the slopes, with the primary broadleaf evergreen forest growing only on mountain crests. In Xujiaba, the most extensive ecosystem is the primary *Lithocarpus* association which covers 75–80% of the Xujiaba region. The broadleaf evergreen *Lithocarpus* grows as a thick forest with a dark and moist understorey where sunlight rarely penetrates the forest floor [26]. The alternation of wet and dry conditions is the typical climate in the Ailaoshan. The mean annual precipitation is 1799 mm, 86% of which occurs in the wet season from May to October [27].

### 2.2. Experimental Design and Sample Collection

The soil used in the study was collected from a field in the Ailao mountains in August 2019 (hereafter 2019Wet), April (hereafter 2020Dry), and August 2020 (hereafter 2020Wet) (Appendix A). Detailed sampling information is provided in Appendix A. Three distant plots were selected in the sampling site ~100 m^2^ in size. Four individual adult trees were randomly selected from each plot, and trees were separated by 5–15 m. In the *Lithocarpus* forest, the ground covers heavy humic fragmented litter (hereafter HF). Firstly, we collected HF and stored them in the bag. Then, we collected the soil below the HF. The soil cores (~20 cm) comprising organic soil (hereafter OS) and organic mineral (hereafter OM) were collected at ~1 m (north and south) from the trunk of the adult tree. OS and OM were collected from the top 10 cm of the soil surface and 10–25 cm below the soil surface. The organic and mineral horizons were studied separately because fungal and bacterial communities were previously found to differ between these horizons due to the differences in nutrient availability and the presence of root-associated microorganisms [28,29]. To remove the roots and other impurities, the soil samples were sieved through 5 mm sterile meshes. They were homogenized and immediately subsampled in sterile plastic tubes to be used for DNA extraction. All of the soil samples were transported to the laboratory on dry ice and stored at −80 °C before DNA extraction.

### 2.3. DNA Extraction, PCR Amplification, and Sequencing

The total DNA was extracted from the soil using the DNeasy PowerSoil kit (Qiagen, Hilden, Germany) according to the manufacturer’s protocol. The DNA quality and quantity were measured with the NanoDrop ONE spectrophotometer (Thermo Scientific, Waltham, MA, USA). For each microbiome determination, the DNA of three replicates was pooled to form one sample. The bacterial and fungal communities were profiled based on the 16S rRNA gene primers: 515F (5′-GTGCCAGCMGCCGCGGTAA-3′) and 806R (5′-GGACTACHVGGGTWTCTAAT-3′), and internal transcribed spacer (ITS) primers: ITS1F (5′-CTTGGTCATTTAGAGGAAGTAA-3′) and ITS2 (5′-GCTGCGTTCTTCATCGATG-3′), with barcodes at the 5′ end of each forward primer [30,31]. All of the PCR reactions were carried out in 30 µL reactions with 15 µL of Phusion^®^ high-Fidelity PCR Master Mix (New England Biolabs, Ipswich, MA, USA), 0.2 µM of forward and reverse primers, and about 10 ng of template DNA. The PCR amplifications were carried out using the following cycling conditions: 1 min initial denaturation at 98 °C, 30 cycles of 10 s at 98 °C, 30 s at 50 °C, and 30 s at 72 °C, with a final 5 min elongation at 72 °C. Libraries were generated using the Illumina TruSeq DNA PCR-Free Library Preparation Kit (Illumina, San Diego, CA, USA) following the manufacturer’s recommendations, and index codes were added. The library quality was assessed on the Qubit 2.0 Fluorometer (Thermo Scientific, Waltham, MA, USA) and Agilent Bioanalyzer 2100 system. Finally, the library was sequenced on an Illumina NovaSeq platform at Novogene Biotech Co., Ltd. (Beijing, China), and 250 bp paired-end reads were generated.

### 2.4. Processing and Analyses of the Sequencing Data

The amplicon sequence data were processed with the DADA2 pipeline as described previously [9,32]. In brief, the raw sequences were split according to their unique barcodes, and the adapter, primer, and barcode were subsequently removed. After their removal, the DADA2 packages were used to process and construct the amplicon sequence variants (ASV) table. We visualized quality and filter reads using the following parameters (maxN = 0, maxEE = c(2, 3) for 16S and maxEE = c(3,5) for ITS). At the same time, the chimeric sequences were removed. Taxonomic assignments for the clustered ASVs were performed using the RDP trainset 16 database for bacterial and the UNITE v2020 database version for fungal ASVs. Overall, paired-end sequencing resulted in 15,644,467 and 16,788,534 high-quality reads from 216 samples, and these reads could be assembled into 5993 and 1390 ASVs for bacteria and fungi, respectively. The phyloseq package was used for downstream analysis of the ASVs table [33]. Non-bacterial ASVs (chloroplast and mitochondrial) were then removed. We also filtered the ASVs which were not annotated at the phylum level. In this study, we found that the non-annotated OTUs have a low abundance (<0.00001), and the results were not different for filtration of non-annotated OTUs and no filtration of non-annotated OTUs. The raw sequencing data were submitted to the Sequence Read Archive under accession number PRJNA782391.

### 2.5. Statistical Analysis and Visualization

All the statistical analyses of the data were performed in the R platform with the use of different packages. Microbial alpha diversity analysis (Shannon and observed ASVs) was calculated as implemented in the R packages phyloseq [33]. The beta diversity was estimated according to the Bray–Curtis dissimilarity distance between the samples. Different factors on the community dissimilarity were tested using PERMANOVA for Bray–Curtis indices with 1000 permutations, as implemented in the adonis function of the R vegan package [9]. A linear mixed model was used to check the effect of the soil layer and sampling season on bacterial and fungal alpha diversity [34]. We used SourceTracker (v1.0.1, Dan Knights, California, CA, USA) software to study the exchange percentages of the soil layer. The Source Tracker analysis was constructed as follows: based on ASVs data, we estimated the proportion of HF and OS communities from OM soil, HF and OM communities from OS soil, and OS and OM communities from HF soil. The percentage value was derived from the statistical average of the results from SourceTracker [35].

To elucidate the microbial interactions between the soil layers and/or sampling seasons, respectively, microbial association networks for each soil layer or sampling season were assessed with a network meta-matrix created using the ASVs table, with the ASVs as rows and samples as columns. To reduce network complexity, the ASVs presented in different percentages of all samples were filtered (only ASVs that were detected to be present in 95 and 20% of all samples for bacteria and fungi network analysis). Firstly, the meta-matrix was generated using the R package “SpiecEasi”, which uses LASSO regularization and cross-validation to detect the most parsimonious network structure in high-dimensional microbial data [4,36]. The lambda ratio was 0.01, and the network was assessed over 20 values of lambda for each 50 cross-validation permutations to detect the least variable network links by the StARS selection criterion [37]. The networks were estimated at each permutation by the glasso graph estimation method. The visualization of the networks and calculation of network topological properties (degree, modularity, and so on) were performed using the interactive platform Gephi [38]. The ASVs with high degree and closeness centrality values were identified as ‘hub species’ in co-occurrence networks. The cutoffs for the hub nodes were set at the degree >30 for bacterial and >20 for fungal, and for closeness centrality >0.3 as hub nodes [34].

We employed the differential abundance analysis to identify the microbial taxa responsible for the community differentiation among the soil layers or sampling season. The analysis was performed using the EdgeR’s generalized linear model (GLM) approach [39]. The differential ASVs with false discovery rate-corrected *p* values < 0.05 were identified as indicator ASVs, which were illustrated by ternary plots with the “ggtern” package [40]. In this study, we defined the dominant taxa (ASVs present in at least 80% of samples for bacterial microbiome or 50% of samples for fungal microbiome and with a relative abundance of ≥0.2% or ≥0.3% for bacterial and fungal microbiome). The phylogenetic tree was annotated and visualized with the iTOL software (https://itol.embl.de/, Last accessed on 18 January 2022) [41]. Linear discriminant analysis effect size (LEfSe) was applied (Wilcoxon *p*-value < 0.05, logarithmic LDA score >1; http://huttenhower.sph.harvard.edu/galaxy/, Last accessed on 18 January 2022) to identify the biomarker of each soil layer and sampling season [42]. A non-parametric statistical test was used to evaluate the taxonomical difference observed in the different soil layers and sampling seasons. All of the statistical tests performed in this study were considered significant at *p* < 0.05.

## 3. Results

### 3.1. Soil Microbiome Assembly Was Most Strongly Influenced by the Soil Layer

The alpha-diversity (Shannon and observed ASVs) analysis of all the bacterial samples showed that diversity in the HF was significantly higher than in OS and OM (*p* < 0.05). Notably, a significant difference was identified between OS and OM for the bacterial microbiome diversity (*p* < 0.05, Figure 1A). Additionally, the 2019Wet season exhibited the highest alpha diversity among the bacterial microbiomes (Figure 1A). Furthermore, the linear mixed model analysis showed that the bacterial richness was mainly influenced by the season (F_2,205_ = 131.77, *p* < 2.2 × 10^−16^), while the bacterial diversity by both the soil layer (F_2,205_ = 74.24, *p* < 2.2 × 10^−16^) and season (F_2,205_ = 63.48, *p* < 2.2 × 10^−16^) (Appendix A). For the fungal microbiome, we found that the alpha diversity of HF was significantly higher than OS and OM (*p* < 0.05, Figure 1A). However, the season when the samples were taken had a significant impact on the fungal microbiome. The alpha diversity of the wet season was significantly higher than that of the dry season (*p* < 0.05). The linear mixed model analysis based on all samples showed that fungal diversity and richness were mainly influenced by the season (F_2,205_ = 29.31, *p* = 6.37 × 10^−12^; F_2,205_ = 32.33, *p* = 6.28 × 10^−13^) (Appendix A).

Non-metric multidimensional scaling (NMDS) analysis showed that HF clearly separated with OS and OM samples and PERMANOVA analysis of complete samples indicated that the variation within the bacterial and fungal communities was mainly explained by the soil layer (*R*^2^ = 34.73%, *p* < 0.001; *R*^2^ = 3.57%, *p* < 0.001) and then by the season (*R*^2^ = 4.95%, *p* < 0.001; *R*^2^ = 2.55%, *p* < 0.001) (Figure 1B and Appendix A). Meanwhile, the soil layer together with season significantly affected bacterial community (R^2^ = 8.6%, *p* < 0.001). Moreover, hierarchical clustering analysis of the fungal and bacterial microbiome revealed a clear difference between soil layers. Most of the HF samples clustered together and were separated from OS and OM in the bacterial microbiome (Appendix A). The HF samples formed a single big cluster, and OS and OM samples clustered together in the fungal microbiome (Appendix A).

### 3.2. Soil Layers Possessed Different Ecological Network Complexity

Network analysis was performed to assess the co-occurrence patterns of the bacterial and fungal communities among the successive soil layers. The average degree was used as a proxy of the network complexity. The results suggested that the soil microbial network complexity declined strongly with soil depth, with the highest microbial network connectivity for the fungal community observed in HF and the lowest connectivity in the OM. On the other hand, bacterial network complexity decreased from OS (with an average degree of 11.34) to HF (10.05) and OM (8.73) (Figure 2 and Table 1). Moreover, the number of ‘hub nodes’ (nodes with high values of degree (>30 and >20 for bacterial and fungal microbiome) and closeness centrality (>0.3) in the network) gradually decreased from OS to HF and OM in the bacterial community and from HF to OS and OM in the fungal community (Table 1).

The taxonomic composition of the network in the fungal microbiome differed between the soil layers, with more nodes belonging to Ascomycota in HF and Basidiomycota in OS and OM. In terms of the bacterial microbiome, the OS (60.14%) and OM (62.16%) possessed a higher percentage of nodes that were annotated as Acidobacteria when compared to the HF (36.04%). In addition, higher modularity was identified in the OS and OM, while the average path distance was found in OM and HF for bacterial and fungal microbiomes (Table 1). To further characterize the seasonal effect on the bacterial and fungal microbiomes, we assessed the alpha-diversity and co-occurrence patterns within the same soil layer. Our results suggested that the season strongly impacted fungal diversity (observed ASVs and Shannon index) and network complexity. The wet season’s fungal diversity and network complexity were higher than the dry season. The 2019Wet season exhibited higher diversity and network complexity in the bacterial microbiome than the 2020Dry season. On the other hand, the 2020Wet season showed lower diversity and network complexity than the 2020Dry season, expect the network complexity of 2020Wet of the OS soil layer and the alpha diversity of HF based on the samples from different sampling seasons (Appendix A).

### 3.3. The Community Composition and Selection Process of Soil Layer

The bacterial and fungal community composition in the different soil layers and sampling seasons are shown in Figure 3. Overall, all soil layers and sampling seasons were dominated by Acidobacteria (52.9 and 52.8%), Actinobacteria (11.3 and 11.3%), and Proteobacteria (26.7 and 26.7%) for bacterial phyla and Agaricomycetes (83.9 and 83.6%) in terms of fungal classes. Notably, the relative abundances of these phyla showed significant differences across different soil compartments and sampling seasons (Figure 3A,B,F,G). OS and OM exhibited higher similarity in bacterial and fungal community composition compared to the HF soil layer. Upon closer inspection of the differences between the soil layers, the HF exhibited higher abundance in Actinobacteria, Bacteroidetes, Proteobacteria, Planctomyctes, Armatimonadetes, and WPS-2 than any other soil layer. In comparison, Acidobacteria and Verrucomicrobia were observed in a higher proportion in OS and OM than in HF (Appendix A, *p* < 0.05). Correspondingly, significant variation in the fungal community composition was also observed between them. Eurotiomycetes, Sordariomycetes, Pezizomycetes, unidentified, and Rozellomycotina_cls_Incertae_sedis were the phyla identified in the different soil layers. The OS and OM contained more Rozellomycotina_cls_Incertae_sedis than HF, while HF contained more Eurotiomycetes, Sordariomycetes, Pezizomycetes, and unidentified compared to OS and OM (Appendix A, *p* < 0.05). We also examined the sampling season effect on the community composition. Notably, the relative abundance of Planctomycetes and Verrucomicrobia during the wet season was significantly higher than that observed during the dry season (Appendix A). When assessing the whole fungal microbiome, there were no significant differences between the dry and wet seasons (Appendix A).

Furthermore, we used the SourceTracker program to identify the exchange proportion of the bacterial and fungal communities among soil layers. Based on the source apportionment results, the exchange proportion of the bacterial microbiome is higher than that of the fungal microbiome (Figure 3C). Assessing the bacterial microbiome, the OS and OM harbored similar patterns. The results indicated that most OS and OM soil bacterial community members (92.55% and 94.72%) were derived from the OM and OS. The exchange proportion between HF and OS was lower compared between OS and OM, respectively, indicating that other environmental sources might contribute to the HF microbiome. By contrast, the exchange proportion of the fungal microbiome is 8.09% (with OS as the source) and 6.58% (with OM as the source) for OM and OS, respectively. Importantly, we found that the exchange proportion from the bottom to the topsoil was smaller than the HF to OS to OM, with a similar pattern observed in the fungal microbiome (Figure 3C). 

To better understand how the soil layer influenced the bacterial and fungal community composition, we identified ASVs enriched explicitly in the soil layer. The HF bacterial microbiome was characterized by many HF-enriched ASVs, mostly belonging to Acidobacteria, Actinobacteria, and Proteobacteria (122 red circles, Figure 3D). Many enriched ASVs (187 red circles), mostly belonging to Ascomycota and Basidiomycota, were found in the fungal microbiome of HF. In contrast, only a few ASVs were found specifically enriched in OS and OM samples (15 green circles and 10 blue circles). All of them were annotated as Acidobacteria and Proteobacteria. When compared to the bacterial microbiome, the fungal microbiome OS- and OM-enriched ASVs (one green circle and one blue circle) were dramatically decreased. Furthermore, we identified the enriched ASVs to further characterize the bacterial and fungal community shift under different sampling seasons. Assessing the bacterial microbiome, the 2019Wet season had the highest enriched ASVs (38) compared to the other sampling season. Enriched ASVs were also identified in similar abundances in the 2020Dry (26) and 2020Wet (27) seasons (Figure 3E). Conversely, in the fungal microbiome, the 2020Dry possessed the lowest enriched ASVs (58) compared to the 2020Dry (69) and 2019Wet (84) seasons, respectively (Figure 3E). 

### 3.4. Soil Microbiome Dominant Taxa and Biomarker among Soil Layers

To further characterize the soil layer effect, we identified the dominant taxa (ASVs present in at least 80% (bacteria) and 50% (fungi) of samples and with a relative abundance >0.2%) and biomarker taxa for each soil layer. Among all of the ASVs obtained from the soil layers, only 116 (2%), 112 (2%), and 115 (2%) ASVs were identified as the dominant taxa for the HF, OS, and OM bacterial microbiomes, respectively. These ASVs accounted for 43% (35.4, 45.6, and 49.1%) of the total sequences in each soil layer. In all soil layers, the dominant ASVs were mainly Acidobacteria, with a relative abundance ranging from 42.6 to 69.5%. There were 38 dominant taxa shared between the different soil layers, with 24 of them annotated as Acidobacteria (Figure 4A–C and Appendix A). Assessing the fungal microbiome, 68 (5.2%), 43 (3.2%), and 48 (3.6%) ASVs were identified as the dominant taxa for HF, OS, and OM, respectively. These ASVs accounted for 57% (46.3, 60.4, and 65.6%) of the total sequences from each soil layer. These dominant ASVs were Agaricomycetes, with a relative abundance of 77–88.5% within each soil layer. The soil layers shared 15 dominant taxa, and nine of them were annotated as Agaricomyceta, belonging to the Russulaceae family (Figure 5 and Appendix A).

We also assessed the sampling season effect. The results indicated that 69 (1.2%), 60 (1%), and 85 (1.5%) ASVs were identified as the dominant taxa for the 2020Dry, 2019Wet, and 2020Wet season bacterial microbiomes, respectively. These ASVs accounted for 31% (31.7, 29.3, and 33%) of the total sequences from each sampling season. These dominant ASVs were mainly Acidobacteria, with a relative abundance of 56.2–60.9% within each sampling season (Appendix A). There were 45 dominant taxa shared by the different sampling seasons, with 27 annotated as Acidobacteria and the remaining annotated as Actinobacteria, Proteobacteria, and Verrucomicrobia at the phylum level (Appendix A). Moreover, when examining the fungal microbiome, 34 (2.7%), 64 (5%), and 79 (6%), ASVs were identified as the dominant taxa for the 2020Dry, 2019Wet, and 2020Wet seasons, respectively. These ASVs accounted for 58% (53.1, 50.1, and 70.3%) of the total sequences from each sampling season. These dominant fungal microbiome ASVs were mainly Agaricomycetes, with a relative abundance of 84.2–88.2% within each sampling season (Appendix A). There were 12 dominant taxa shared among the different sampling seasons, with six of them annotated as Agaricomyceta (Appendix A).

The LDA effect size (LEfSe) analysis uncovered Acidobacteriaceae in HF and uncultured bacteria belonging to the phylum of Acidobacteria in OS and OM as the most significant bacterial biomarker taxa. In the fungal microbiome, Hydnodontaceae in HF, *Nigrospora* in OS, and Archaeorhizomycetaceae in OM were identified as the most significant biomarker taxa (Appendix A). The dominant taxa for the different sampling seasons were also identified. Burkholderiaceae for 2020Dry, Acidobacteriales for 2019Wet, and Verrucomicrobia for 2020Wet season were the most significant biomarker taxa. In the fungal microbiome, the most significant biomarker taxa were Tremellales for 2020Dry, Leotiomycetes for 2019Wet, and Cortinariaceae for 2020Wet season, respectively (Appendix A).

## 4. Discussion

In the present study, we examined the diversity and composition of the soil layer under different sampling seasons of the *Lithocarpus* association forest. Our results showed that HF’s bacterial and fungal richness and diversity were higher than in the OS and OM. The soil layer primarily shapes the microbiome, with marginal influence from sampling season. We further discovered that the soil layer strongly affected network complexity. Moreover, we identified each soil layer’s composition, exchange proportion, and dominant taxa through multiple machine-learning methods, which provided more information for further study of the ecosystem function of the soil microbiome.

### 4.1. The Assembly Pattern and Divergence of Soil Layer Microbiome

Our results showed that bacterial diversity and richness reached the highest values in the HF layer and the lowest in the OM layer. Specifically, the fungal diversity and richness were the highest in HF and the lowest in the OS soil layer (Figure 1A). Furthermore, we identified that the microbiome assembly is primarily determined by the soil layer rather than by sampling season (Figure 1B). These results suggested that each soil layer harbored a divergent microbial composition. This is in line with the published results in which the alpha diversity of the soil layer displayed a decreasing trend with depth and principal coordinate analysis of the soil layer clearly separated [17,18,43]. Although the richness and diversity of OS were lower than the OM for the fungal microbiome, it did not display significant differences (Figure 1A). The vertical differences in the soil layer could be attributed to the decline in the availability of various resources with soil depth [44]. The linear model analysis indicated that the diversity and richness were mainly influenced by sampling season (Appendix A). Consistent with previous studies, the results demonstrated that seasonal changes play an essential role in the bacterial and fungal microbiome [45,46]. The wet season increases the fungal microbiome’s diversity for each soil layer compared with the dry season. On the other hand, the bacteria microbiome’s diversity did not follow the same consistency concerning the sampling season (Appendix A).

The soil structure and moisture content influence the creation of microbial habitats and niches with cascading effects on carbon and nutrient transformations. Therefore, understanding microbial connectivity is required to understand better how it affects species interactions [47,48]. The co-occurrence network analysis further indicated that the fungal microbiome’s complexity decreased with the soil depth. At the same time, the complexity of the bacteria microbiome displayed that the highest complexity network was observed in the OS and the lowest in the OM soil layer (Figure 2). Furthermore, the interactions in the wet season were more complex than those in the dry season for the fungal microbiome. However, for the bacterial microbiome, there was no similar pattern for different soil layers (Appendix A). Yet, we identified that the season influenced the fungal communities more than the bacterial communities. Previous studies show that drought increased the connectedness and centrality of nodes in bacterial networks while decreasing these properties in fungal networks [49]. In terms of the bacterial microbiome, we displayed that the network complexity of 2019Wet is higher than the 2020Dry season for each soil layer (Appendix A).

### 4.2. The Community Composition and Selection Process of Soil Layer

Our community composition results displayed that the HF composition differed from OS and OM, with OS and OM exhibiting similar compositions (Figure 3). Correspondingly, the relative abundances of Actinobacteria, Bacteroidetes, Proteobacteria, Planctomycetes, and Armatimonadetes were more abundant in HF, while the Acidobacteria and Verrucomicrobia were deleted, respectively. Only Rozellomycota differed between the soil layer in the fungal microbiome (Appendix A). The compositions were similar to the published paper [18]. Notably, the relative abundance of Others (the relative abundance <0.5%) from the wet season is higher than in the dry season for the fungal microbiome. For the bacterial microbiome, the relative abundance of Planctomycetes and Verrucomicrobia in the wet season was significantly higher than in the dry season (Appendix A). These results demonstrated that the rare taxa are more sensitive to environmental factors than the dominant taxa. This is in line with the research papers that showed that the rare taxa are more sensitive to host selection and play an essential role in the fungal co-occurrence network and ecosystem functioning [12].

A vastly diverse microbiota generally colonizes plants. However, the structure, abundance, and occurrence of microorganisms can be variable across different plant habitats [50,51,52]. Soil habitats are the significant sources of crop microbial selection, and specific taxa are gradually enriched while others are filtered out by different host niches [34]. Soil habitats are the major, and other external forces, such as wind, rainfall, and crawling insects, also contribute to it [53]. Our results showed that the HF was enriched with more bacterial and fungal-specific ASVs than the OS and OM (Figure 3D). The upper soil possesses unique OTUs than the other strata based on the OTU composition analysis [18]. Meanwhile, the source-tracking and composition analysis revealed that the OS and OM are more similar compared to HF (Figure 3). HF is mainly composed of the litter of forest trees, and all of the results demonstrated that HF possessed different microbiomes compared with the soil microbiome. Our results are in line with the previous research in which the microbiome analysis in roots and above-ground compartments of poplar trees and sugarcane suggested that different root or leaf niches harbored distinct microbial communities [11,54].

Previous studies have shown that the bulk soil is the primary source of microbial species richness in plant rhizosphere, and crop-associated bacteria are derived primarily from bulk soils [34,55]. Importantly, we found that the exchange proportion of the fungal microbiome dramatically decreased compared with the bacterial microbiome (Figure 3). Such differences between bacteria and fungi, the two dominant components of the soil microbiome, likely reflect the general characteristics of the different dispersal behaviors of these two classes of organisms [56,57]. The fungal distribution exhibits strong biogeographic patterns that could be driven by dispersal limitations, while bacteria are shown to have weak biogeographical patterns [58]. As the oxygen decreases with soil depth, it can strongly influence microbial composition since the fungal and bacterial microbiomes prefer different oxygen conditions [17,59]. The above results show that the soil layer is an important factor affecting microbial diversity and structure.

### 4.3. The Dominant Taxa for Each Soil Layer

The network hubs, dominant taxa, and biomarker taxa were considered potential keystone taxa that have an essential ecological role in microbiome assembly and ecosystem functions [60,61]. Our results suggested that 2% and 4% of bacterial and fungal ASVs consistently accounted for 43% and 57% of the bacterial and fungal microbiome in the soil layer, indicating that only a few microbial taxa dominate in different soil layers. This is in line with the previous research showing that only 2% of bacterial phylotypes accounted for half of the soil bacterial communities [60]. Our results indicated that Acidobacteria and Agaricomycetes were the most dominant taxa in the bacterial and fungal microbiome, respectively (Figure 4 and Figure 5). Acidobacteria represents an enigmatic phylum with members copiously distributed in different ecosystems. Acidobacteria performs specific ecological functions such as regulating biogeochemical cycles and decomposing biopolymers. As the forest ecosystem contains enormous reservoirs of dead trees and litter, the decomposition of this material and mobilization of nutrients are essential for forest health. Similarly, Agaricomycetes play a crucial role in cycling nutrients in forest soils [62,63]. This firmly explains their dominant distribution in the soil layer.

Our results showed that most hub nodes of HF were annotated to Ascomycota when compared to OS and OM, respectively. Ascomycota plays a vital role in degrading organic matter, and its content may impact soil fertility [16,64]. In the meantime, we found that the Hydnondontaceae, which belongs to Agaricomycetes, is the biomarker taxa in the HF. As mentioned above, Agaricomycetes are responsible for lignocellulose decomposition. Similarly, the most significant taxa of OS and OM were Nigrospora and Archaeorhizomycetaceae, all of which belong to Ascomycota, respectively. These results illustrate the differences between the soil layers. The significant bacterial microbiome biomarker in the soil layers was the Acidobacteria phylum (Appendix A). Similarly, Acidobacteria are the hub node for the soil layers, which only display the different percentages. This could explain its distributed features, which are distributed in nearly all ecosystems [62].

## Figures and Tables

**Figure 1 jof-08-00948-f001:**
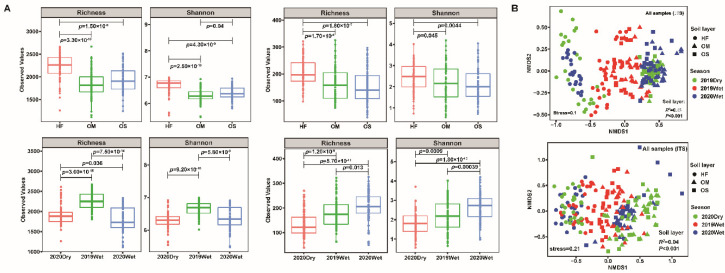
Soil layer effects on the soil microbiome. (**A**) Comparative analysis of the alpha diversity of the bacterial (left) and fungal (right) community. (**B**) Non-metric multidimensional scaling (NMDS) ordinations based on Bray–Curtis distance of bacterial (up) and fungal (down) community composition across all samples. HF: humic fragmented litter, OS: organic soil, OM: organic mineral soil. 2019Wet: samples collected from wet season in 2019; 2020Dry and 2020Wet: samples collected from dry season and wet season in 2020.

**Figure 2 jof-08-00948-f002:**
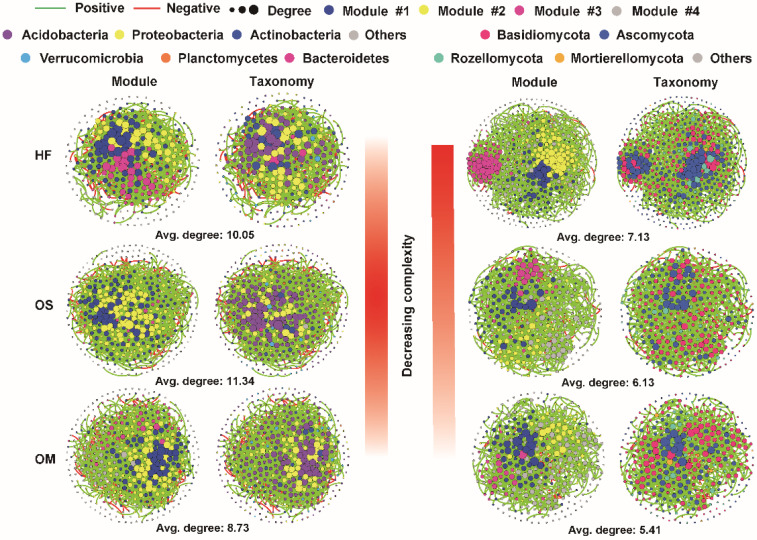
Bacterial and fungal co-occurrence network along soil layers based on all samples. For visual clarity, only amplicon sequence variants (ASVs) that were detected to be present in 95% and 20% of all samples for bacteria and fungi are illustrated. HF: humic fragmented litter, OS: organic soil, OM: organic mineral soil.

**Figure 3 jof-08-00948-f003:**
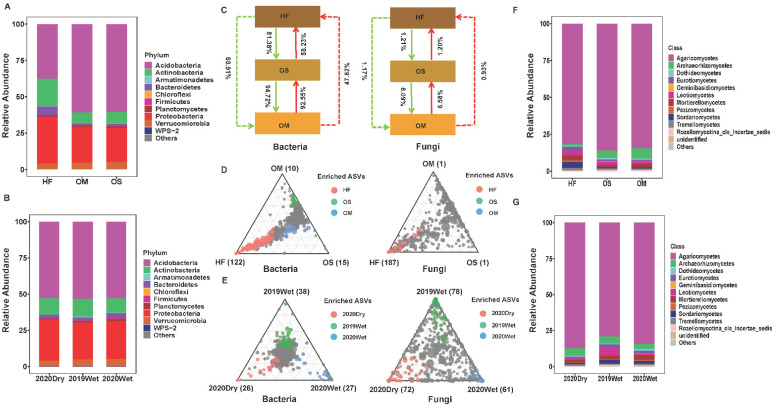
The community composition and selection process of the soil layers. The relative abundance of the most abundant of bacterial phyla (**A**) and fungal (**F**) class among soil layers. The relative abundance of the most abundant of bacterial (**B**) and fungal (**G**) phyla among sampling seasons. (**C**) Source model showing the potential exchange based on all samples. Bacterial and fungal ASVs shared among soil layer (**D**) and sampling season (**E**). Ternary plots depict the relative abundance of all ASVs (>0.5%) for soil layer or sampling season across bacterial and fungal microbiomes. Each point corresponds to an ASV. Its position represents its relative abundance for each soil layer, and its size represents the average across all soil layers. Colored circles represent ASVs enriched in one soil layer compared to the others (red in HF or 2020Dry, green in OS or 2019Wet, and blue in OM or 2020Wet). The phyla with less than 0.05% of the average relative abundance are grouped into “Other”. HF: humic fragmented litter, OS: organic soil, OM: organic mineral soil. ASVs: amplicon sequence variants. 2019Wet: samples collected from wet season in 2019; 2020Dry and 2020Wet: samples collected from dry season and wet season in 2020.

**Figure 4 jof-08-00948-f004:**
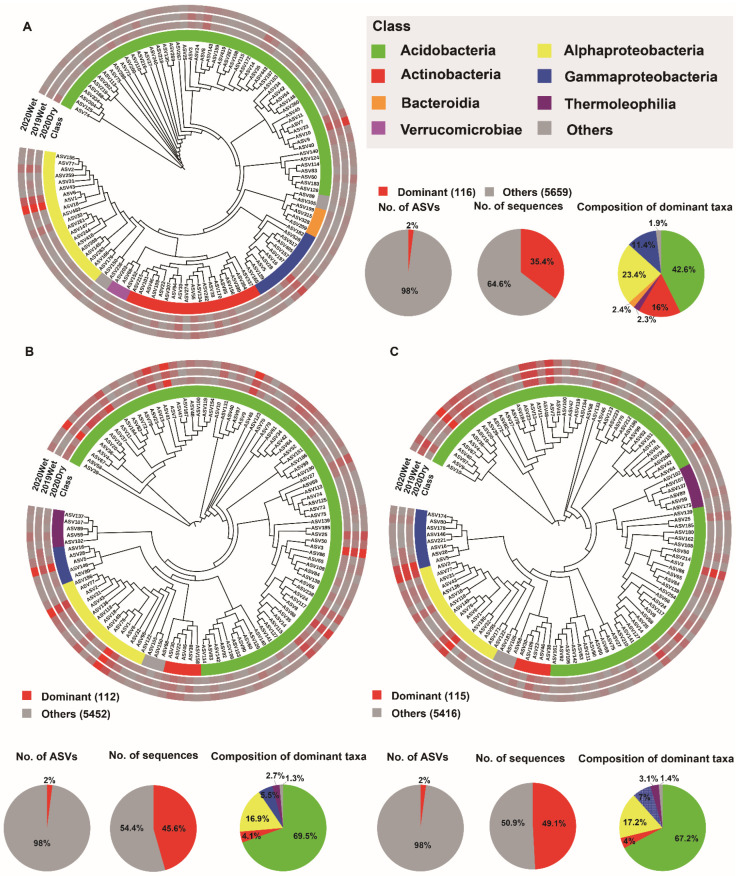
Phylogenetic tree, taxonomic composition, and distribution patterns of soil layer bacterial microbiome dominant taxa. (**A**) Identification of dominant taxa in HF (n = 72). (**B**) Identification of dominant taxa in OS (n = 72). (**C**) Identification of dominant taxa in OM (n = 72). The dominant taxa were defined as ASVs present in more than 80% of all samples and with an average relative abundance ≥0.2%. Low abundance classes with <2% of the total sequences of dominant taxa across soil layer are grouped into ‘Others’. HF: humic fragmented litter, OS: organic soil, OM: organic mineral soil. ASVs: amplicon sequence variants. 2019Wet: samples collected from wet season in 2019; 2020Dry and 2020Wet: samples collected from dry season and wet season in 2020.

**Figure 5 jof-08-00948-f005:**
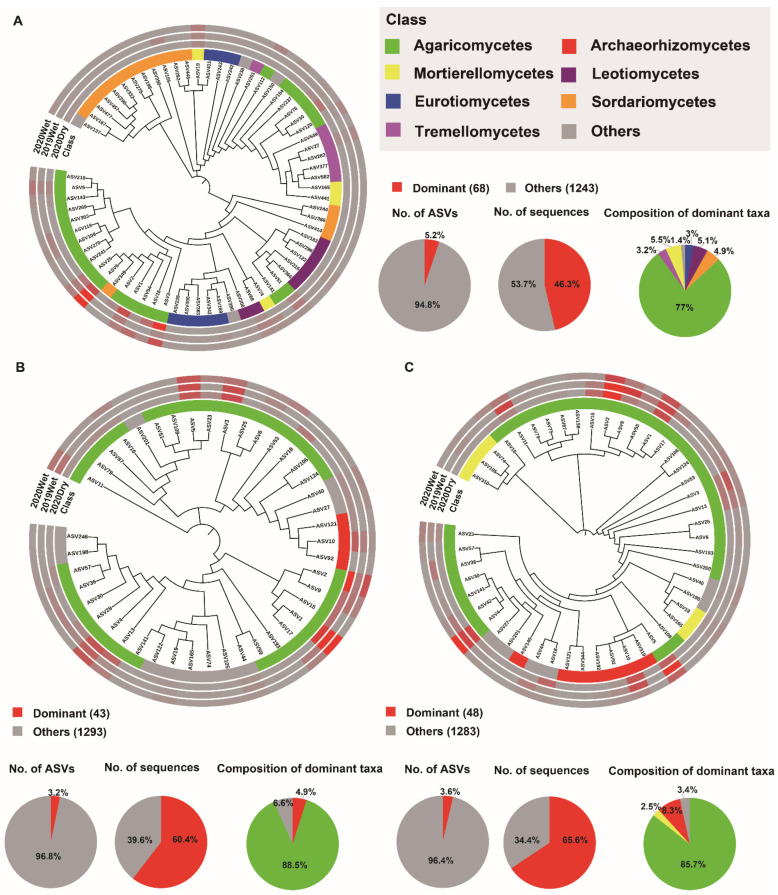
Phylogenetic tree, taxonomic composition, and distribution patterns of soil layer fungal microbiome dominant taxa. (**A**) Identification of dominant taxa in HF (n = 72). (**B**) Identification of dominant taxa in OS (n = 72). (**C**) Identification of dominant taxa in OM (n = 72). The dominant taxa were defined as ASVs present in more than 50% of all samples and with an average relative abundance ≥0.2%. Low abundance classes with <2% of the total sequences of dominant taxa across soil layer are grouped into ‘Others’. HF: humic fragmented litter, OS: organic soil, OM: organic mineral soil. ASVs: amplicon sequence variants. 2019Wet: samples collected from wet season in 2019; 2020Dry and 2020Wet: samples collected from dry season and wet season in 2020.

**Table 1 jof-08-00948-t001:** Bacterial and fungal co-occurrence network characteristics in each soil layers.

	Soil Layer	Node	Positive Edge	Negative Edge	Average Degree	Modularity	Average Clustering Coefficient	Average Path Distance	Hub Node
Bacteria	HF	308	977	571	10.05	1.561	0.454	3.115	23
	OS	439	1503	985	11.34	1.905	0.335	3.143	43
	OM	444	1246	691	8.73	1.449	0.324	3.361	20
Fungi	HF	512	1605	220	7.13	0.815	0.355	4.063	24
	OS	357	994	100	6.13	0.799	0.349	3.902	6
	OM	353	857	97	5.41	0.82	0.364	3.903	5

HF: humic fragemented litter, OS: organic soil, OM: organic mineral soil.

## Data Availability

The raw sequencing data have been submitted to the Sequence Read Archive under accession number PRJNA782391.

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
