# Peer review of "Soil Layers Impact Lithocarpus Soil Microbial Composition in the Ailao Mountains Subtropical Forest, Yunnan, China"

_jof, 2022, doi:10.3390/jof8090948_

Round 1

Reviewer 1 Report

The manuscript describes the soil structure impact on soil microbial composition. The work is interesting, but I feel that some clarifications should be made prior to publication. Bellow you can find my suggestions.

Minor:

Also, the authors need to use scientific language. This type of wording is not adequate for a scientific publication: “Firstly, we collected humic fragments using a knife and stored them in a big bag.” (lines117-118)

English should be revised through the manuscript. These are some examples, but are not an exhaustive list:

lines 28-27, sentence is not complete.

Lines 116-117, please correct: “In the Lithocarpus forest, it covers heavy humic fragemented litter”

Lines211: correct the sentence “…which assessed by Shannon and observed ASVs index,…”

Abstract, lane 25-26. Which environmental factors?

Which is the objective of this work? It should be clearly stated in the abstract section.

Introduction

Line 49. “Plants are divided into annual, biennial and perennial plants”. In fact, plants there are numerous ways to cluster plants. The one indicated by the author is just one. I advise rephrasing.

Lines 65-66: “Soil represents the most important habitat for the soil microbiome." This is obvious and useless.

Lines 80-81: “Previous studies have indicated that bacteria and fungi predominate in the early stages and latter stages of litter decomposition” Please clarify.

Objective: the authors define the investigations objectives as: “(i) identify the microbiome composition of the soil layer and the seasonal variation. (ii) determine the relative contribution of the environmental factors and soil layers in shaping microbiome assemblies; (iii) assess the co-occurrence patterns across soil layers and identify the potential source and keystone taxa of the different soil layers.” The authors should address the gaps in knowledge that make these objective relevant: i) Why is seasonal variation is going to be addressed? ii)Which environmental factors are going to be studied. And why?

Also, authors spend a significant amount of time describing the relevance of litterfall in the abstract. But it is not considered in the introduction.

Materials and methods

Not all material/methods are defined or explained. As an example: why have sample locations been named as wet and dry? Is it related to pluviosity? Do the authors have those data?

Results

How was the enrichment analysis (fig3) made?

Discussion

The authors need to clarify why they believe that “The soil layer primarily shapes the microbiome, with marginal influence from environmental factors” since no environmental factors were considered. On the other hand, the authors also state that “The liner model analysis indicated that the diversity and richness were mainly influenced by season” which seems to contradict the previous sentence.

Figures and tables:

 - define all abbreviations (eg. HF, OS, OM); what are T1-T4, ….? ASV,

- complete figure and table captions. All figures/tables should stand on their own. The reader should be able to understand the figure/table without going to the text. Give all the details necessary to understand data on table/figures.

Reviewer 2 Report

I reviewed the article "Soil structure impact on Lithocarpus soil microbial composition at the Ailao Mountains subtropical forest". The article covers an important aspect of soil science and the authors have produced a high amount the data. However, the article needs to be improved before publication. I present below details on the suggested modifications that could guide authors to improve the manuscript.

1- TITLE: it should be modified. You are not evaluating soil structure impact on microbial composition but impacts of soil layers and seasons.

2- ABSTRACT: it should be modified after incorporation of major changes in analyses and results.

3- INTRODUCTION: Introduction is not bad, but I found it too long and losing the focus in some paragraphs. You don't need to go deeper in each aspect. You need to present the research gaps and the objectives of your research. You could do that with one or two simple sentences instead of several paragraphs. Moreover, authors need to support the statements. Just to provide you with some examples:

- L39-40: you need to add at least one reference here.

- Line 42: you need to add at least one reference after "Most of Earth's living biomass is comprised of microorganisms".

- Line 45: you need to add at least one reference after "The microorganisms colonizing the soil are the most abundant and diverse life forms on Earth".

- Lines 47-54: try to focus. Also the final sentence is too ambitious. When reading reader think on a study covering all plant microbiomes and this is not what you are covering. The same is valid for others final paragraph sentences. Just to focus on research gaps and the goals of your research.

-Lines 74-75: you need to add at least one reference after "Cover crops have been widely applied globally as a conservation agriculture practice".

You have a lot of cases like these, where references should be added.

AIMS: Please you need to revise this part: 

- you are focus only on microbiome composition, when in fact you are analysing diversity and composition.

- environmental factors were not evaluated in the manuscript. The only factors considered in the study were "soil layer" and "sampling season". You could not consider "sampling season" as synonym of "environmental factors". You are not testing impacts of environmental factors.

- on network analysis it is not clear why you focus on soil layer when your results indicate that both factors "soil layer" and "sampling season" have significant impacts on microbial communities.

Globally in the manuscript be careful on using "litter" as synonym of "soil layer".

MATERIAL AND METHODS

- Line 114, is "plots" equivalent to "clusters" as indicated in Figure S1. If yes, please select one term and use it through the manuscript and supplementary files.

- Line 120: please explain why you are sampling in north and south directions only (meaning why you are not also sampling east and west points)

- Lines 161-162: why you are filtering the not annotated ASVs? not annotated ASVs are also important. We could not assign taxonomy to those ASVs but they could be also relevant for the soil. Did you tested if relative abundance of non-annotated ASVs changed among soil layers and seasons?  They are also part of soil communities. Taxonomy assignation is limited by the available sequences with taxonomy on the databases but an ASV without taxonomy is not meaning that the ASV is not relevant and should be excluded. The constraints in the taxonomy assignation are in the databases, not in the soil. You need to include non-annotated ASVs in you analyses.

- Line 167: please indicate what diversity index you used for the study.

- Lines 168-169: please indicate what type of distance matrix you used (similarity or dissimilarity).

- Lines 171-172: do not generalize with "major drivers", you are not considering a lot of drivers, you are just considering two factors "soil layer" and "sampling season". So I suggest indicate clearly what factors where considered in the analyses and to reorganize the sentence... you are not identifying the major drivers, you are evaluating the effects of "soil layer" and "sampling season" on microbial alfa-diversity.

-Lines 178-193: why are you considered separately the analysis of networks for "soil layer" and "sampling season" when your results indicate that there is significant interaction between both factors. This is not clear and should be explained.

- Lines 181-182: This is not clear. Could you please clearly the criteria used to filter ASVs? I am not sure that filtering ASVs having different relative abundance among samples is a good criterion. How did you know that this ASVs are not relevant for the network? Not clear. 

- Lines 194- 207: as before not clear why you are performing these analyses separately for "soil layer" and "sampling season" when you previously showed the significant interaction of both factors.

RESULTS

- Lines 210- 223: Why are you presenting results of each factor separately if you perform diversity analysis combining "soil layer" AND "sampling season" and considering the interaction. The main result here is that both factors and the interaction are significant for bacterial and fungi as you show in TableS2.
You can go deeper then to see the specific differences but you need to show both factors in the same figure. There is no reason (even is not correct) to present one graph for soil layer and one graph for season if the interaction was significant. You need to generate the combined graph and present pairwise differences in the graph or in a supplementary table.

Also, in order to organize the paragraph, I suggest to describe bacterial and fungi separately. And do not use terms as "wet season" you have two different wet seasons and there are differences between them. So generalizing results using "wet season" is not correct.
You can discuss differences between bacterial and fungal communities in the discussion section. However, here in the Results I strongly suggest to present bacterial and fungal communities separately, for clarity.

Lines 225-239: NMDS analysis is not described in M&M sections. Please added it. And please, be careful with the presentation of the analyses. NDMS is an explorative multivariate analysis, effect of different factors is not tested in this method. Please present results of NDMS and PERMANOVA separately. Moreover, I suggest to think why you are performing both analyses. If you want to test effects of soil layer and season on microbial composition you need to used PERMANOVA. If you want to explore how samples separate based on multivariate data, you should perform NDMS.
If you are interested in both types of information you should perform both analyses but you need to present results separately because the goal of each type of analysis is different.

Also you have a significant interaction between "soil layer" and "sampling season" according to TableS3, why are you presenting each factor separately, you need to consider both factors together and the interaction. These R2 values correspond to what? This is not correct. You have a significant interaction so you need to explore pairwise differences in the PERMANOVA.

Figure 1: You need to revise the legend. In figure 1A bacterial is in the left and fungi on the right, not up and down... and you are considering not only soil layers but also seasons. In figure 1C you need to clarify which graphs correspond to bacteria and fungi.

For the rest of the results section, I will not go deeper because I need first you clarify why you are analysing factors separately and why/how you are filtering ASVs. Also as I previously indicate I consider that not-annotated ASVs should be not removed for the analyses.

I am not providing comments for the Discussion section as I consider that results will change when introducing suggested modifications. But I recommend authors to be focus in this section.

I hope my comments guide the authors to improve the manuscript.  

Round 2

Reviewer 1 Report

The manuscript was improved but there are still some alterations that should be made prior to publication.

My main concern relates to the definition of the objective. The authors should clearly state which is the objective. Stating a gap in knowledge "the microbiome composition and the interaction of litterfall and soil remain poor understand" is not the same as identifying the objective.

Reviewer 2 Report

I reviewed the article "Soil layers impact on Lithocarpus soil microbial composition at 2 the Ailao Mountains subtropical forest". Although you make some improvements in the manuscript I still considered that the manuscript is not ready for publication.

For example, analyses are still presented by factors when interaction between factors was significant. I did not agree with this decision because a lot of important information is missed.

In the same way the filtration of non-annotated ASVs was maintained, although authors recognize that this is not a good decision because non-annotated ASVs are also important.

Moreover I did not agree with the justification that authors indicate in the response, argumenting that they are performing analyses in the same way that other published research. This is not a correct argument. You need to clearly present the criteria for your analyses and be able to sustain your decision. Citation of other articles is valid to support information or discussion but you need to define your criteria clearly.

As I mentioned in my first review, you have a lot of data and results but you need to work on the presentation. Working with each factor separately is not recommended.

I suggest authors to work on the manuscript considering the recommendations of the previous review.

Round 3

Reviewer 1 Report

Thank you for your efforts to improve the manuscript.

Reviewer 2 Report

I finished my third review of the manuscript "Soil layers impact on Lithocarpus soil microbial composition at the Ailao Mountains subtropical forest". Unfortunatelly the manuscript still have several issues in analyses and presentation of the data. 

Authors are not taking comments and suggestions into consideration, they are justifying in the cover letter some points. However, they are not using this exchange to improve the manuscript. Just to provide an example, authors included a good justification about why unannoted ASVs where excluded from the analyses, but they did not include this justification in the manuscript neither indicated the low frequency of unannotated ASVs in the text.

More important, authors arbitrarily decided to present results ignoring that the interaction of both factors "soil layer" and "seasons" was significant almost for all analyses. I strongly disagree at this point as I mentioned in all reviews, because authors are ignoring an importan result of the study. Even considering factors separately results should be better interpreted. For example, almost for all analyses significant differences were observed between HF and OS/OM, but OS and OM were not differenciated. Therefore conclusions presented in Discussion section are not correct. Just to mention one example: "Our results showed that bacterial diversity and richness decreased with increasing soil depth". This sentence is not true because differences were not significant between OS and OM wit the exception of fungal diversity. Similar results were obtained for beta diversity and other analyses, with the exception of network analysis. Differences were mostly and generally between HF and OS/OM. In a similar way results on season should be considered carefully and I think in interaction with soil depth.

Therefore, although considering that the authors have good data, I still consider that they should improve analyses and presentation of the results, and that they need to base discussion and conclusions strictly on the results.

After this third review I consider that the manuscript should be rejected.

I encourage authors to take comments and considerations to improve the manuscript.
